# The Safety and Efficacy of Glucosamine and/or Chondroitin in Humans: A Systematic Review

**DOI:** 10.3390/nu17132093

**Published:** 2025-06-24

**Authors:** Kyrie Eleyson R. Baden, Sarah L. Hoeksema, Nathan Gibson, Divine N. Gadi, Eliya Craig, Juanita A. Draime, Stephanie M. Tubb, Aleda M. H. Chen

**Affiliations:** School of Pharmacy, Cedarville University, Cedarville, OH 45314, USA; shoeksema@cedarville.edu (S.L.H.); ngibson@cedarville.edu (N.G.); divinegadi@cedarville.edu (D.N.G.); ecraig@cedarville.edu (E.C.); juanitaadraime@cedarville.edu (J.A.D.); smtubb@cedarville.edu (S.M.T.); amchen@cedarville.edu (A.M.H.C.)

**Keywords:** glucosamine, chondroitin, human

## Abstract

Background/Objectives: Glucosamine and chondroitin are natural substances often used alone or in combination for conditions affecting the joints. Our objective was to evaluate the efficacy and safety of glucosamine and/or chondroitin supplementation in humans as well as to determine the common dosages used. Methods: A systematic review was conducted using PRISMA methodology. Searches were performed in PubMed and Web of Science and uploaded into Covidence where two independent researchers reviewed articles according to inclusion and exclusion criteria. Quality assessment was performed using the Mixed Methods Appraisal Tool (MMAT). Results: Of the 2013 articles screened, 146 studies were included in our review, with nearly 60% being randomized controlled trials and most conducted in Europe, Asia, or the U.S. Most studies focused on osteoarthritis and joint pain, with over 90% of efficacy studies reporting positive outcomes and most safety studies indicating minimal or no adverse effects. Glucosamine and chondroitin were most commonly administered together at daily doses of 1500 mg and 1200 mg, respectively, and often compared to a placebo or celecoxib. Conclusions: Overall, the evidence suggests that glucosamine and chondroitin are generally effective and well-tolerated, particularly for managing osteoarthritis and joint pain. Consistent dosing strategies and favorable safety profiles across a diverse range of studies support their continued use in clinical practice, but further research is needed related to other disease states.

## 1. Introduction

Joint dysfunction is a common cause of morbidity that can arise in various forms. Osteoarthritis (OA) occurs due to degeneration of the cartilage and tissues in the joint primarily due to aging, joint injury or overuse, and obesity [1]. OA, especially of the knee joint, continues to increase around the world, affecting an estimated 528 million people in 2019 [1]. In addition, 73% of these individuals are 55 years or older, and the knee tends to be the most frequently affected joint [1]. Temporomandibular disorder (TMD) is another common group of disorders that affects the joints and muscles surrounding the jaw. The prevalence of TMD increases from adolescence (11%) to adulthood (31%) and is associated with psychological and physical health conditions, such as stress, trouble sleeping, migraines, and chronic fatigue syndrome [2]. Rheumatoid arthritis (RA), on the other hand, is a much less common condition of the joint. This inflammatory, autoimmune disease affects 18 million worldwide, with 70% being women and 55% being 55 years or older [3]. With all of these conditions, joint pain may be an accompanying symptom, or it may be a solitary complication of its own.

Glucosamine and chondroitin (GC) are both components of the extracellular matrix of articular cartilage. It is thought that the mechanisms by which GC work in OA include anti-inflammation, the prevention of cartilage degradation, and the regulation of anabolic processes [4]. Glucosamine has been shown to decrease inflammation by the reduction in several different inflammatory mediators, including reactive oxygen species, nuclear factor κB (NF-κB) activation, C-reactive protein (CRP) interleukin (IL)-1, IL-6, and tumor necrosis factor (TNF)-α, while upregulating the anti-inflammatory mediators IL-2 and IL-10 [5,6]. Chondroitin has been shown to reduce inflammation primarily through the reduction in (NF-κB) activation and in IL-1b, which is a key inflammatory marker that causes articular inflammation and cartilage damage [4,7]. 

In addition, GC both works to prevent cartilage degradation and promote anabolic effects. Not only does anti-inflammation play a role in protecting cartilage but the inhibition of catabolic enzymes can also contribute to this effect. Glucosamine has been shown to inhibit phospholipase A2, matrix metalloproteinases (MMPs), and aggregases, whereas chondroitin has been shown to significantly decrease collagenolytic activity and to induce proteoglycan production [4,7]. Glucosamine has also been shown to increase aggrecan and collagen type II, which are components of the extracellular matrix [8]. Because of these mechanisms, it is thought that these two nutritional supplements work together synergistically in managing OA and other conditions affecting joints and cartilage.

To evaluate the efficacy of various interventions on joint health, several questionnaires are commonly used. Of these, the Western Ontario and McMaster Universities Osteoarthritis Index (WOMAC) [9] is one of the most widely administered. The WOMAC is a validated, 24-item, patient-reported instrument that assesses pain, stiffness, and physical function in knee or hip OA. Similarly, the Lequesne Index (LI) [10] and Lequesne Functional Index (LFI) [11] assess the severity of hip OA via a questionnaire on pain, maximum distance walked, and activities of daily living [10]. Other common instruments that are not disease-specific include the Visual Analog Scale (VAS), which allows a patient to subjectively rate pain, and the Short Form-36 (SF-36), which allows a patient to report quality of life. 

Current guidelines for the treatment of OA recommend both nonpharmacological (physical, psychosocial, and mind-body) and/or pharmacological approaches [12]. Pharmacological treatment includes oral and topical nonsteroidal anti-inflammatory drugs (NSAIDs), acetaminophen, intra-articular corticosteroids, tramadol, duloxetine, and capsaicin [12,13]. NSAIDs are often recommended first-line in OA due to their ability to reduce inflammation and pain; however, gastrointestinal toxicity and nephrotoxicity can occur, as well as higher rates of cardiovascular events [13]. While the topical application of these medications reduces adverse events, topical use is not recommended in polyarticular OA, widespread pain, or hip arthritis [13]. The initial recommended treatment for TMD, on the other hand, includes supportive patient education, cognitive behavioral therapy, and the use of NSAIDs. Tricyclic antidepressants and muscle relaxants are also used. Referral to an oral and maxillofacial surgeon is recommended if conservative therapy is ineffective [14].

It is clear that additional approaches to the management of joint dysfunction, particularly OA, are needed to target underlying causes for progression. Glucosamine and chondroitin, both separately and in combination, have been studied in joint dysfunction with promising outcomes. However, there are still mixed results in terms of efficacy and safety for their use. Therefore, we conducted a systematic review of the literature to evaluate the efficacy, safety, and common dosages of glucosamine and/or chondroitin in humans with any joint or health condition.

## 2. Materials and Methods

A systematic review of the literature was conducted following the PRISMA [15] checklist. The following search terms were used: (“Glucosamine” [Mesh]) AND “Chondroitin” [Mesh]. To ensure appropriateness, a research librarian reviewed the search process and terms before study commencement. Searches were performed in PubMed and Web of Science and uploaded into Covidence, a systematic review management system. The inclusion criteria included (1) studies published from 1 January 1990 up to the date of our search (5 September 2024), (2) research articles (randomized control trial, clinical trial, case study), (3) studies available in full-text, (4) studies available in English, and (5) studies assessing glucosamine/chondroitin efficacy, safety, and/or dosing in humans. The exclusion criteria included studies published outside of our specified time frame, non-research publications (such as commentaries), systematic reviews, meta-analyses, studies without full-text available, studies not available in English, and studies on species other than humans (e.g., animal-based studies). These inclusion criteria were used throughout the post-search review process. In addition, meta-analyses/systematic reviews identified were used to screen for missed articles but were ultimately excluded to limit methodological redundancy.

After Covidence automatically removed duplicate articles, researchers screened articles according to the inclusion and exclusion criteria for each phase: title, abstract screening, and full-text review. Two independent researchers had to agree on the final decision before the article could move on to the next phase. Any conflicts were resolved by a third senior researcher who verified if a given study aligned with our specified criteria. For data extraction, a template was used in Covidence to ensure consistency and included the study characteristics, medications used and doses, conditions, and study outcomes (efficacy, safety). Quality assessment was performed using the Mixed Methods Appraisal Tool (MMAT) [16]. All studies were evaluated on whether or not they had a clear research question and whether the collected data addressed the research question (2 items). The remaining 5 items assessed quality markers associated with the study design. Possible scores could range from 0 to 7, with higher points indicating greater quality.

One researcher performed data extraction and quality assessment, with a senior researcher verifying accuracy. After completion, the research team synthesized data in accordance with the study objectives until a consensus was achieved.

## 3. Results

After the removal of duplicates, 2013 articles were screened for eligibility. Of these articles, 155 were included for data extraction. At the end of the extraction, 9 additional duplicate articles were removed, so a total of 146 studies were included in our review. The PRISMA flow diagram in Figure 1 depicts how articles were included or excluded in each phase. 

### Study Characteristics

Of the 146 studies included [17,18,19,20,21,22,23,24,25,26,27,28,29,30,31,32,33,34,35,36,37,38,39,40,41,42,43,44,45,46,47,48,49,50,51,52,53,54,55,56,57,58,59,60,61,62,63,64,65,66,67,68,69,70,71,72,73,74,75,76,77,78,79,80,81,82,83,84,85,86,87,88,89,90,91,92,93,94,95,96,97,98,99,100,101,102,103,104,105,106,107,108,109,110,111,112,113,114,115,116,117,118,119,120,121,122,123,124,125,126,127,128,129,130,131,132,133,134,135,136,137,138,139,140,141,142,143,144,145,146,147,148,149,150,151,152,153,154,155,156,157,158,159,160,161,162], 59.6% were randomized controlled trials (RCTs). The majority of articles were conducted in Europe (34.9%), Asia (24.0%), and the United States (20.5%), and most studies included both glucosamine and chondroitin (61.6%) rather than glucosamine only or chondroitin only. OA was the primary condition studied (65.1%), with joint pain being the second (13.7%). Other conditions studied for efficacy purposes included TMD, RA, Kashin-Beck disease (KBD), and fibromyalgia. For safety, several conditions that were evaluated to be caused by glucosamine and/or chondroitin included acute liver injury or liver disease, kidney disease/function, allergic reaction, mortality and cardiovascular events (acute myocardial infarction, acute coronary events, ischemic stroke), knee replacement or knee injury, various cancers, dementia, heart failure, type 2 diabetes, inflammation, back pain, heparin-induced thrombocytopenia (HIT), intraocular pressure and glaucoma, and urinary tract infection. Table 1 provides a summary of the key study characteristics.

An overview of study outcomes can be seen in Table 2. Of the 113 studies that evaluated efficacy in a certain condition, 102 reported positive changes in their efficacy parameters. Of the 107 studies that reviewed safety, 80 reported that glucosamine and/or chondroitin had either no adverse effects (AEs), minimal/mild AEs, or AEs no different than the comparator. Five case series/reports of AEs related to GC were included in this safety evaluation. 

Of the 110 studies that specified dosing, 57 reported 1500 mg as the daily dose for glucosamine; however, doses ranged from 0.2 mg (for intervertebral injection) to 3200 mg. In addition, 37 of these studies reported 1200 mg as the daily dose for chondroitin, with doses ranging from 0.005 mg (for intervertebral injection) to 2400 mg.

Appendix A provide details of each individual study included in our review. Appendix A provides further information on study characteristics, including the glucosamine and/or chondroitin dose as well as its comparator (if any), the condition(s) studied, and measures of assessing efficacy or safety. Appendix A provide detailed information on study outcomes for efficacy and safety, respectively. Of the studies evaluating OA and joint pain, the primary measures used to assess efficacy were the Western Ontario and McMaster Universities Arthritis Index (WOMAC) (n = 56), the visual analog scale (VAS) (n = 55), and the Lequesne Index (LI) and Lequesne Functional Index (LFI) (n = 21 total). Patient self-assessments and quality of life questionnaires were also commonly used. The primary biomarkers used to assess efficacy included various markers of inflammation, such as C-reactive protein (CRP), erythrocyte sedimentation rate (ESR), interleukins (ILs), and tumor necrosis factor alpha (TNF-α) or cartilage/bone degradation markers, such as C-terminal crosslinking telopeptide of type II collagen (CTX-II), and matrix metalloproteinase (MMP).

Across all of the studies, there were >4,000,000 participants. There was a total of 15,152 participants across the 87 RCTs included. For the RCTs, the median study duration was 24 weeks (with a mean of 30 weeks). In addition, dosing strategies were similar among the RCTs. The most frequent daily dose of glucosamine was 1500 mg (n = 45 RCTs) with a range of 10 mg to 2250 mg daily. For chondroitin, the most frequent daily dose was 1200 mg (n = 30 RCTs) with a range of 2.5 mg to 1600 mg. Most studies gave both glucosamine and chondroitin in two or three divided doses. In addition, other than placebo, the most common comparator of efficacy was celecoxib 200 mg daily. Overall, across all studies (both RCT and non-RCT), the most common dose of 1500 mg/1200 mg divided three times daily of glucosamine/chondroitin remained consistent.

Table 3 shows the quality assessment of all the included studies. All of the studies had clear research questions and collected data to address the research questions. The average quality rating was 6.4, with 140 of 146 (96%) having a score of ≥6.

## 4. Discussion

Glucosamine and/or chondroitin demonstrated efficacy with minimal safety concerns in most of our included studies. These supplements were shown to be most beneficial in OA and joint pain, whereas efficacy in rheumatoid arthritis was not evident. The measures of efficacy were both subjective (i.e., assessed by the patient with a scale such as the Western Ontario and McMaster Universities Osteoarthritis Index (WOMAC), Lequesne Index (LI), or Visual Analogue Scale (VAS)) and objective (i.e., with markers of inflammation or radiographical imaging of the joint space). As mentioned previously, most studies indicated that glucosamine and/or chondroitin demonstrated efficacy on one or more parameters in OA and joint pain. Several of the studies that showed no benefit included those using glucosamine as a monotherapy [77,125,151] or the dose was not specified [153], indicating that the combination of both supplements may be optimal in the treatment of OA. Furthermore, all studies evaluating GC in TMD showed benefit in one or more efficacy parameters [18,30,39,43,51,107,134]. This pattern was also seen in articles studying KBD [152,155,158]. However, the studies that assessed GC use in RA did not seem to be favorable in some or all tested parameters [91,101]. 

Of note, glucosamine and chondroitin were often used in combination with other therapies, including type II collagen peptides, s-adenosylmethionine (SAMe), curcuminoids, Boswellia, methylsulfonylmethane (MSM), various vitamins, quercetin glycosides, hyaluronan or hyaluronic acid, omega-3 fatty acids, and different forms of exercise. Type II collagen peptides stood out as a consistently efficacious component either in combination or compared to GC [22,41,69,70,86,87,119,120,146]. However, GC also showed a benefit on its own. Therefore, using GC with other treatments may be more specific to the patient and their condition. 

Furthermore, the anti-inflammatory and cartilage-preserving properties of GC were evident in our review. While most studies compared glucosamine and/or chondroitin to placebo, beneficial effects of GC were still seen when compared to celecoxib and diclofenac. In addition, inflammatory mediators were reduced in many studies after use of GC, including CRP [65,66,104,127,159], IL-1ß and IL-6 [43,51,159], and ESR [127,159] and TNF-α [159]. Together, these findings support the theory that GC reduces inflammation. Additionally, many studies also found that GC prevented or reduced the rate of cartilage breakdown, reduced joint space narrowing, reduced markers of cartilage breakdown (e.g., MMP-3), and improved maximal mouth opening (MMO) in TMD. These objective findings, combined with the improved subjective findings on patient satisfaction and quality of life in many studies, demonstrate that GC can provide many advantages for joint conditions.

Based on our review and from the previously established literature, the standard oral dosing strategy of GC is 1500 mg/1200 mg divided into two or three doses per day. While topical application appeared to be more beneficial when compared to placebo in one study [38], it did not demonstrate a benefit over physical therapy in another study [47]. Therefore, oral formulations of GC should be recommended for increased efficacy. Since plasma levels of GC correlate well with synovial levels [113], a greater effect would be expected with higher doses. On the other hand, a smaller effect would be expected with lower doses of GC or with concurrent administration of acetaminophen [59]. This is a particularly interesting finding as acetaminophen is a commonly used medication in the treatment of OA. Nevertheless, a network meta-analysis by Zhu et al. showed that the efficacy of acetaminophen is limited in OA [163]; thus, there is no need to combine this treatment with GC. Further research is needed in this area [24].

Regarding safety, most studies reported no side effects or minimal/mild AEs. The most common AEs reported were gastrointestinal-related, including nausea, diarrhea, constipation, dyspepsia, and bloating. A few studies also reported increased instances of infections [82,150] and allergic reactions [36,41,56,58] and one reported lower eGFR [34]. Rare AEs reported in case series or case reports were drug-induced cholestatic jaundice [63], autoimmune hepatitis [145], and elevated transaminases [32]. In addition, a few studies reported instances of an increased risk for effects on the cardiovascular system/cardiovascular events [37,82,154] or cancer [67,82]. However, other studies reported no increased risk for these complications [26,27]. Finally, intraocular pressure was an AE noted in two studies [48,80]. Interestingly, several protective effects were also reported as GC was associated with a decreased risk of dementia in two different studies [160,162]. For comparison of the side effects, NSAIDs and celecoxib (common treatments for OA and TMD) are known to cause gastrointestinal side effects and also substantially increase cardiovascular risk, which was evident throughout the studies comparing GC to these agents.

Our review may provide a new insight that differs from current guidelines on the use of GC in OA. Currently, OA guidelines from the Arthritis Foundation and American College of Rheumatology conditionally recommend chondroitin sulfate for patients with hand OA but recommend against GC alone or in combination for hip and knee OA [12]. Similarly, the American Academy of Orthopaedic Surgeons (AAOSs) states that GC may be helpful for some patients with mild-to-moderate knee OA [164]; however, no strong recommendation is made for its use. In contrast, 89% (58 of 65) of studies in our review evaluating these supplements in hand, knee, or hip OA reported efficacy in at least one measure. While knee OA represented 6 of the 7 studies that found no benefit in OA, it is important to note that many studies did show efficacy in all forms of OA. Therefore, the use of GC in OA may be more beneficial than previously thought. This review has several marked strengths. Our systematic review followed the PRISMA framework, which reinforced the robustness of our methodology. Furthermore, this review provided an update on GC efficacy and safety from the year 1990 to the present time, allowing for a more complete and current understanding of the literature. Moreover, our MMAT rating demonstrated that nearly all our studies were of high quality. The limitations of our study include the consistency in the researcher’s application of our inclusion and exclusion criteria in each phase of the screening process. However, this was mitigated by the approval of the senior investigator before an article moved on to the next phase. In addition, the heterogeneity of study designs and methods limits the broad interpretation of the data. Furthermore, most studies included in this review evaluated GC in OA and joint pain. Thus, the application of this information outside these conditions, such as in RA, is limited.

Because our research was primarily centered around OA and joint pain, future studies are needed for further evaluation in other disease states. In addition, more research is needed to assess how the efficacy, safety, and optimal dosing of GC differs based on patient age, sex, and comorbidities. This will allow for more patient-specific recommendations of glucosamine and/or chondroitin. 

## 5. Conclusions

Glucosamine and chondroitin provide beneficial effects on efficacy, particularly in osteoarthritis and joint pain. These effects seemed especially positive when GC was used in combination rather than alone, suggesting these two ingredients have a synergistic relationship. The benefit of GC remained consistent when compared to an active control, such as celecoxib. However, further benefit may be seen with other nutraceuticals or nutritional supplements. In addition, GC alone and in combination have minimal safety concerns, with the primary side effects being mild, gastrointestinal complaints. Our results support the oral daily dose of 1500 mg and 1200 mg for glucosamine and chondroitin, respectively, in two or three divided doses. Future studies are needed to determine optimal dosing based on other patient-specific factors.

## Figures and Tables

**Figure 1 nutrients-17-02093-f001:**
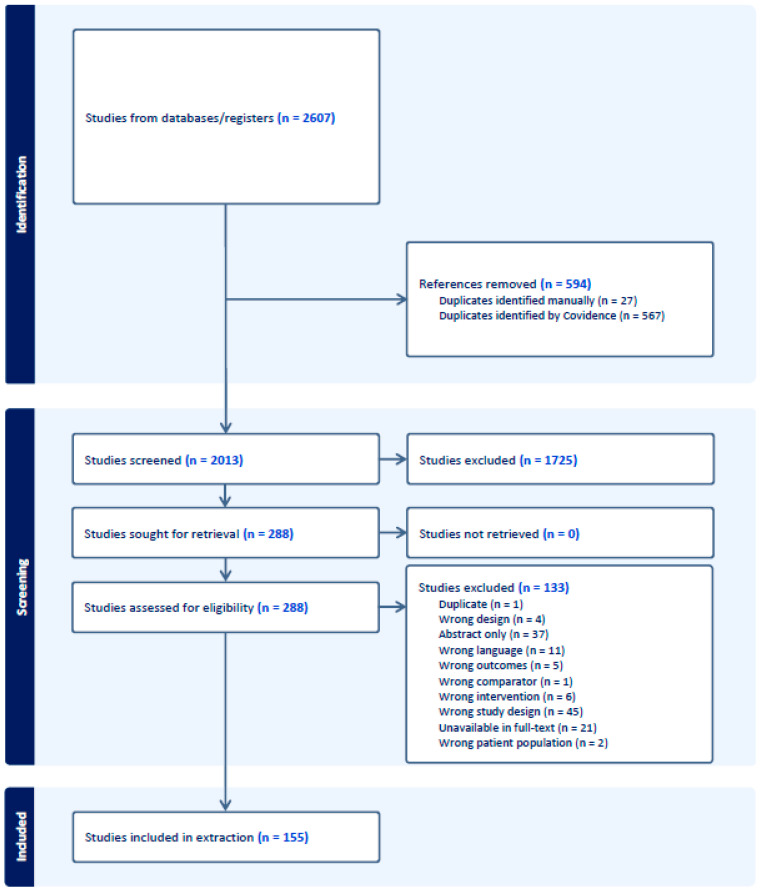
PRISMA diagram overviewing the study inclusion and exclusion process.

**Table 1 nutrients-17-02093-t001:** Condensed study characteristics for studies evaluating glucosamine and/or chondroitin.

Study Characteristic	n of Studies (%)
Country	
Asia	35 (24.0)
Australia	4 (2.7)
Canada	10 (6.8)
Europe	51 (34.9)
Middle East	12 (8.2)
South/Central America	4 (2.7)
United States	30 (20.5)
Study Design	
RCT	87 (59.6)
Non-RCT experimental	12 (8.2)
Cohort	19 (13.0)
Cross-Sectional	6 (4.1)
Case-Control	6 (4.1)
Case series/report	6 (4.1)
Other	10 (6.8)
Medication	
Glucosamine only	42 (28.8)
Chondroitin only	14 (9.6)
Glucosamine + chondroitin	90 (61.6)
Condition	
Osteoarthritis	95 (65.1)
Joint pain	20 (13.7)
Temporomandibular Joint Dysfunction	7 (4.8)
Rheumatoid arthritis	3 (2.1)
Kashin-Beck disease	3 (2.1)
Fibromyalgia	1 (0.7)
Mortality	3 (2.1)

**Table 2 nutrients-17-02093-t002:** Glucosamine and/or chondroitin efficacy and safety summary findings.

Health Condition	Range of Dosing per Day	Summary of Efficacy Outcomes	Summary of Safety Outcomes
Osteoarthritis	Range glucosamine: 375–2250 mg Range chondroitin: 60–1560 mg	102 of 113 efficacy studies found benefit on efficacy parameters in patients taking glucosamine and/or chondroitin	80 of 107 safety studies reported no adverse effects, minimal/mild adverse effects, or adverse effects no different than the comparator.
Joint Pain	Range glucosamine: 200–1500 mg Range chondroitin: 60–1200 mg		
Temporomandibular Joint Dysfunction	Range glucosamine: 1200–3200 mg Range chondroitin: 1200–2400 mg		
Rheumatoid Arthritis	Range glucosamine: 1200–1500 mg Range chondroitin: 111 mg		
Kashin-Beck Disease	Range glucosamine: 800–1440 mg Range chondroitin: 800–1200 mg		
Other Conditions	Range glucosamine: 0.2–3000 mg Range chondroitin: 0.005–1200 mg		

**Table 3 nutrients-17-02093-t003:** Mixed Methods Assessment Tool (MMAT) quality assessment of the included studies.

Article ID (Author Year)	Mixed Methods Appraisal Tool (MMAT) Composite Score (Possible Range 0–7)
Alayat 2017 [17]	7
Alhayek 2023 [18]	7
Amalraj 2019 [19]	7
Armagan 2015 [20]	7
Arora 2020 [21]	6
Ayhan 2024 [22]	7
Babur 2022 [23]	7
Basak 2004 [24]	6
Belcaro 2014 [25]	7
Bell 2012 [26]	7
Bhimani 2023 [27]	7
Blakeley 2002 [28]	7
Boeri 2024 [29]	6
Cahlin 2011 [30]	7
Catanzaro 2013 [31]	7
Cerda 2013 [32]	7
Cho 2019 [33]	7
Cho 2023 [34]	7
Chopra 2013 [35]	7
Chu 2023 [36]	7
Clegg 2006 [37]	7
Cohen 2003 [38]	7
Cömert Kılıç 2021 [39]	6
Conrozier 2019 [40]	6
Crowley 2009 [41]	7
Czajka 2018 [42]	6
Damlar 2015 [43]	4
Das 2000 [44]	6
Dorais 2018 [45]	6
Eraslan 2015 [46]	6
Erhan 2012 [47]	7
Esfandiari 2017 [48]	6
Filipović 2022 [49]	6
Fransen 2015 [50]	7
Ganti 2018 [51]	6
Giordano 2009 [52]	7
Greenlee 2013 [53]	7
Gruenwald 2009 [54]	6
Herrero-Beaumont 2007 [55]	6
Hoban 2020 [56]	7
Hochberg 2008 [57]	6
Hochberg 2016 [58]	7
Hoffer 2001 [59]	5
Hotaling 2011 [60]	7
Hsu 2019 [61]	7
Ibáñez-Sanz 2020 [62]	7
Ip 2015 [63]	7
Issa 2021 [64]	6
Kantor 2012 [65]	6
Kantor 2014 [66]	7
Kantor 2016 [67]	7
Kanzaki 2012 [68]	6
Kanzaki 2015 [69]	6
Kanzaki 2016 [70]	5
Kawasaki 2008 [71]	7
Khanna 2020 [72]	5
King 2020 [73]	7
Klein 2003 [74]	6
Kongtharvonskul 2016 [75]	7
Kubový 2012 [76]	7
Kwoh 2014 [77]	6
Lapane 2012 [78]	6
Leffler 1999 [79]	6
Lehrer 2024 [80]	6
Li 2023 [81]	7
Lila 2023 [82]	6
Lomonte 2018 [83]	6
Lomonte 2021 [84]	6
Lubis 2017 [85]	7
Lugo 2016 [86]	6
Luo 2022 [87]	5
Ma 2019 [88]	7
Magrans-Courtney 2011 [89]	6
Martel-Pelletier 2017 [90]	6
Matsuno 2009 [91]	7
Mazières 2007 [92]	6
Mazzucchelli 2021 [93]	7
Mazzucchelli 2022 [94]	7
Messier 2007 [95]	6
Michel 2005 [96]	7
Minoretti 2024 [97]	6
Monfort 2017 [98]	6
Morita 2018 [99]	6
Muftic 2024 [100]	3
Nakamura 2007 [101]	6
Nakasone 2011 [102]	6
Nash 2018 [103]	7
Navarro 2015 [104]	7
Navarro 2019 [105]	6
Navarro 2020 [106]	7
Nguyen 2001 [107]	7
Nieman 2013 [108]	7
Pavelká 2002 [109]	6
Pelletier 2016 [110]	7
Peluso 2016 [111]	6
Persiani 2005 [112]	6
Persiani 2007 [113]	7
Petersen 2011 [114]	6
Pocobelli 2010 [115]	7
Pontes 2018 [116]	7
Provenza 2015 [117]	6
Puente 2017 [118]	6
PuigdellívolGrifell 2024 [120]	6
Puigdellivol 2019 [119]	7
Raaijmaakers 2008 [121]	7
Railhac 2012 [122]	6
Raynauld 2016 [123]	7
Reginster 2001 [124]	6
Rindone 2000 [125]	6
Roman-Blas 2017 [126]	7
Rondanelli 2019 [127]	6
Rondanelli 2020 [128]	6
Roubille 2015 [129]	7
Rovati 2016 [130]	7
Sawitzke 2008 [131]	7
Scroggie 2003 [132]	6
Sevimli 2020 [133]	6
Shankland 1998 [134]	6
Sterzi 2016 [135]	6
Thomas 2021 [136]	6
Tí o 2017 [137]	6
Tokhiriyon 2019 [138]	6
Truong 2019 [139]	7
Tsuji 2016 [140]	6
Uebelhart 2004 [141]	6
Usha 2004 [142]	6
Velickovic 2023 [143]	6
Vicenzino 2019 [144]	7
vonFelden 2013 [145]	7
Vreju 2019 [146]	7
Wang 2021 [147]	6
Wang 2021 [148]	6
Weimann 2001 [149]	6
Wildi 2011 [150]	7
Wilkens 2010 [151]	6
Xia 2016 [152]	7
Yang 2015 [153]	7
Yu 2022 [154]	7
Yue 2012 [155]	6
Zegels 2013 [156]	6
Zenk 2002 [157]	6
Zhang 2010 [158]	7
Zhang 2021 [159]	6
Zheng 2023 [160]	7
Zheng 2023 [161]	6
Zhou 2023 [162]	7

## Data Availability

No new data were created outside of what is published in this article. The study protocol, data collection forms, data extracted from included studies, and all other materials used in this review can be made available upon request.

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
