# Peer review of "The Safety and Efficacy of Glucosamine and/or Chondroitin in Humans: A Systematic Review"

_nutrients, 2025, doi:10.3390/nu17132093_

Round 1

Reviewer 1 Report

Comments and Suggestions for Authors

I learned a lot from this article. Thank you.

The Safety and Efficacy of Glucosamine and/or Chondroitin in Humans: A Systematic Review

The manuscript is a well-structured, up-to-date systematic review that synthesises 146 human studies on glucosamine and chondroitin. Your adherence to PRISMA methodology, large timespan (1990-2024) and the consistent application of MMAT quality appraisal make the evidence summary both comprehensive and methodologically sound. The clear finding that >90 % of efficacy studies report benefits while most safety studies report only mild adverse events will be valuable for clinicians and nutrition scientists alike.

  • Abstract: Replace qualitative phrases (“over 90 % reporting positive outcomes”) with quantitative effect sizes where possible (e.g. pooled WOMAC mean difference, risk ratio for adverse events). This will give readers an immediate sense of clinical magnitude.
  • Introduction: Explicitly articulate the PICO question (Population, Intervention, Comparator, Outcomes). This will link seamlessly to the eligibility criteria in Section 2.
  • Results: Provide separate columns for mono-therapy vs combination trials and add median follow-up duration; this will assist readers in judging exposure adequacy.

Reviewer 2 Report

Comments and Suggestions for Authors

The authors nicely present their findings from a systematic review of the efficacy and safety of glucosamine +/- chondroitin supplementation in humans. The manuscript represents an impressive evaluation of the literature. The authors followed established guidelines for performing the review. The findings are clearly presented in tables and figures. The conclusions are appropriate, based on the evidence described.

Minor suggestions:

Include the mixed methods appraisal tool in the supplementary materials.

Consider organizing the list of articles within each study section alphabetically by first author.

Reviewer 3 Report

Comments and Suggestions for Authors

This manuscript presents a comprehensive and clinically relevant systematic review examining the safety, efficacy, and dosing of glucosamine and/or chondroitin in human populations. The review addresses an important topic, particularly for managing joint conditions such as osteoarthritis, and adheres to PRISMA guidelines. The inclusion of a wide range of study designs and outcome measures enhances the relevance and breadth of the findings. The detailed tables summarizing individual study characteristics, interventions, and outcomes are a strength; however, some sections require clarification and refinement to improve the manuscript’s accuracy and clarity.

First, there is an inconsistency in the number of studies reported. On Page 4, Line 122, the manuscript states that 146 studies were included, whereas Figure 1 shows that 155 studies were included in the extraction phase. This discrepancy should be resolved by carefully reviewing the final count and ensuring consistency between the text and figure. In addition, the rationale for excluding systematic reviews and meta-analyses (Page 3, Lines 96–98) should be explained more clearly. A brief justification based on methodological redundancy or lack of original data would improve transparency.

The Results section, while informative, is overly dense. Large tables, such as Table 3, may be better suited as supplementary material to avoid overwhelming readers. A concise summary table or visual figure highlighting key outcomes (e.g., percentage of studies showing efficacy, frequency of adverse effects) would enhance accessibility. Furthermore, although the authors conducted a narrative synthesis, the absence of quantitative analysis (e.g., pooled effect estimates or outcome direction summaries) limits the interpretability of the findings. If a meta-analysis is not feasible, a more structured description of trends across conditions (e.g., osteoarthritis vs. TMD) would strengthen the conclusions. Likewise, defining outcome measures such as WOMAC, VAS, and HAQ earlier in the manuscript would improve clarity for readers unfamiliar with these tools.

In summary, the review is a valuable contribution to the literature, but minor to moderate revisions are required to ensure accuracy, transparency, and reader engagement. Addressing the noted inconsistencies, streamlining the presentation of data, and enhancing the synthesis of findings will greatly improve the manuscript's overall quality and clarity.

Comments on the Quality of English Language

The quality of English throughout the manuscript is generally clear and understandable. The authors use appropriate scientific terminology, and the structure of the sentences is largely sound. However, there are several areas where the clarity, grammar, and flow could be improved with careful editing. Minor grammatical issues, such as inconsistent verb tense, awkward phrasing, and punctuation errors, are present and may affect the readability of some sections.

Reviewer 4 Report

Comments and Suggestions for Authors

Respected Authors,

The thorough and well-written systematic evaluation of clinical research on the effectiveness and safety of glucosamine and/or chondroitin supplementation in humans is what your article is. Nearly 60% of the 146 original papers included in the review were randomized clinical trials. The analysis considers the variety of indications, doses, populations, and results. Positive clinical outcomes and a solid safety profile are demonstrated in the great majority of trials. The article can be a handy resource for researchers and physicians because of the topic's currentness, the breadth of the investigation, and the data's clear presentation. However, it has to be improved in its current form to address the following problems:

  1. Despite the extensive range of study designs (doses, circumstances, duration), the conclusions include unambiguous claims of efficacy and safety. It is essential to highlight the limitations of the data's heterogeneity and provide a more nuanced evaluation (e.g., effects stronger in OA than RA, missing data for some groups).
  2. Although I know the study is not a meta-analysis, it would be tempting to conduct fundamental quantitative analyses, such as calculating the number of RCTs demonstrating efficacy at a specific dosage.
  3. To improve transparency, kindly include an MMAT score summary or a quality heatmap for the included research.

4 There are repetitions, and the article is over 70 pages long. To make it easier for the reader to locate the information of interest, I advise looking for triplicates in the article and attempting to visually portray some of the data (at least the dosages utilized, depending on the condition/disease). Additionally, I suggest condensing tables (such as 3-5) and relocating part of the information to the supplementary material.

  1. Please comment on the review's findings in light of the most recent recommendations for using glucosamine and chondroitin in OA.

Best regards,

The reviewer
